# Characterization of the Root Nodule Microbiome of the Exotic Tree *Falcataria falcata* (Fabaceae) in Guangdong, Southern China

**Siyu Xiang [†], Shu Yan [†], Qianxi Lin, Rong Huang, Runhui Wang, Ruping Wei, Guandi Wu and Huiquan Zheng *** 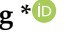

Guangdong Provincial Key Laboratory of Silviculture, Protection and Utilization,
Guangdong Academy of Forestry, Guangzhou 510520, China; xiangsiyu4781@163.com (S.X.);
yanshu@sinogaf.cn (S.Y.); lqx6198@163.com (Q.L.); huangrong@sinogaf.cn (R.H.); wrh@sinogaf.cn (R.W.);
weirp@sinogaf.cn (R.W.); wgd@sinogaf.cn (G.W.)
* Correspondence: zhenghq@sinogaf.cn; Tel.: +86-20-8758-4306; Fax: +86-20-8703-1245
† These authors contributed equally to this work.

**Abstract:** *Falcataria falcata* is an exotic tree species that was imported to southern China around 1940 and has been widely planted in the Guangdong province of China. Using the 16S rRNA amplicon sequencing approach, we investigated the composition of the bacterial endophytes in the root nodules of naturally grown *F. falcata* and elucidated the core bacterial endophyte group. Across all samples, there were 575 bacterial genera and 29 bacterial phyla. Proteobacteria accounted for 42–90% relative abundance in all regions. Notably, *Bradyrhizobium*, *Paucibacter*, *Rhizobium*, and *Mesorhizobium* were consistently detected in all regions studied. Among these, *Bradyrhizobium* (13–37%) and *Paucibacter* (1–34%) were the dominant genera. Despite the differences in endophytic amplicon sequence variants (ASVs) across all regions, our results demonstrate that some ASVs, which have been termed herein as commonly shared core ASVs (c-ASVs), still inhabit *F. falcata* root nodules across multiple regions simultaneously. More importantly, some c-ASVs dominated in *F. falcata* root nodules across multiple regions. This study demonstrated the consistency of the bacterial endophyte communities of *F. falcata* root nodules.

**Keywords:** nodule; microbiome; *Falcataria falcata*; 16S rRNA; *Bradyrhizobium*

## 1. Introduction

As highly capable model bacteria, rhizobia can elucidate the interaction between plants and root microbiota, and in particular how legumes adapt to diverse substrates [1]. Root nodules are formed by the interaction of rhizobia and plants [2]. Among diverse legume species, a remarkable feature is their capacity to harness atmospheric nitrogen ($N_2$) through symbiotic associations with rhizobia residing within specialized root nodules [3]. Rhizobia must survive in soil as free-living organisms before entering plant nodules to survive as nitrogen-fixing symbionts [4]. As part of the rhizosphere microbiota, rhizobia can have a profound impact on the growth, nutrition, and health of legumes [5].

Rhizobia of the genera *Rhizobium*, *Burkholderia*, *Mesorhizobium*, *Bradyrhizobium*, *Ensifer*, *Kaistia*, and *Ochrobactrum* can nodulate plants under diverse natural conditions [1,6–8]. Multiple factors influence host plant selection by particular rhizobial endophytes, including soil pH [9], soil salinity [10], water [11], geographic locations [12], rhizosphere microorganisms [1] and plant genetic links [13]. Soil is a highly heterogeneous environment, and the structure of the soil microbiota varies greatly at centimeter scales [14]. It is unclear whether the same rhizobia are selected for nodulation when plants grow in different natural habitats. Consequently, a research goal is to examine plants' selective function in the relative abundance of rhizobia in various geographic regions.

*Falcataria falcata* (*Fabaceae*, legumes) is an important exotic tree species in southern China utilized extensively for landscape improvement and forestation. Originally native

to the Philippines and Indonesia and introduced to China around 1940, *F. falcata* has a long history of cultivation in these regions, primarily due to its rapid growth rate [15]. Interestingly, *F. falcata* has thrived in its new environment and has maintained its original ecological characteristics, including the formation of root nodules. Similar to most of the other leguminous, *F. falcata* forms symbiotic relationships with specific soil rhizobia and develops root nodules. This mutualistic association benefits the growth of *F. falcata*, and the long-term cultivation of this plant can greatly enhance soil fertility [16]. From *F. falcata* root nodules, *Bradyrhizobium* was isolated in multiple regions of Indonesia [17]. The number of endophytic bacteria that were obtained by isolating them from plants after surface sterilization and culturing them in the laboratory medium may only represent a small fraction of the total endophytic bacteria present in the sample. It is worth mentioning that the endophytic bacteria that have not been successfully isolated are often referred to as viable but nonculturable, and they also play important roles in plants [18,19].

Understanding the decipherable structure of bacterial endophyte communities in *F. falcata* root nodules in southern China is crucial for comprehending the adaptation of *F. falcata* to the foreign environment as an exotic tree species. Other than rhizobia, nonrhizobial root nodule endophytes have been perceived and discovered to possess possible benefits for the host plant's growth [20–22]. The positive effects of endophytic bacteria isolated from nodules have been confirmed through inoculation experiments, which include the promotion of nodule formation [23] and the enhancement of stress resistance [24]. However, inoculation with endophytes alone does not guarantee an automatic increase in plant fitness [25]. The inoculation of suitable endophytes is crucial for producing a positive effect on the plant [24]. This underscores the concept that specific strains of legume rhizobia can establish symbiotic relationships in specific environments [7]. It is vital to identify plants' essential endophytic microorganisms to effectively implement the strategy of "inoculation with specific endophytes".

We expect to learn more in this research about the *F. falcata* root nodules' microbiome and the core bacterial endophyte group across nine represented regions of Guangdong in Southern China using the 16S rRNA amplicon sequencing approach. This is potentially the first exhaustive investigation of bacterial endophytes' structure in the root nodules of *F. falcata* since its inception in southern China.

## 2. Materials and Methods

### 2.1. Nodule Collection

We obtained the root nodules during the fruiting season of *F. falcata*, which is native to the southern Chinese province of Guangdong. The sampling sites were located within a longitude range of 111°14′55″ E to 116°55′28″ E and a latitude range of 21°37′43″ N to 23°45′21″ N (Figure 1 and Table 1). We sampled the root nodules in the north, south, east, and west directions of *F. falcata*, and all the nodules that were collected from one tree were mixed as one sample.

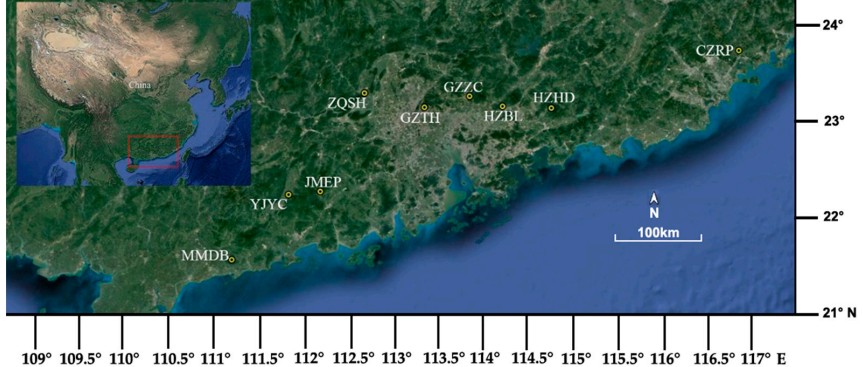

**Figure 1.** Geographic distribution of *Falcataria falcata* root nodule samples. The original satellite image was obtained from Google Map, and modified with Adobe Illustrator 2022.

**Table 1.** The sampling scheme for the root nodules in the experiment.

| Group | Sample | Location | Latitude (° E) | Longitude (° N) | Number of Sampled Trees |
|---|---|---|---|---|---|
| MMDB | MM01, MM02, MM03 | Dianbai, Maoming, Guangdong | 111.2487 | 21.62869 | 3 |
| YJYC | YC01 | Yangjiang, Yangchun, Guangdong | 111.8722 | 22.29952 | 1 |
| JMEP | JM01 | Enping, Jiangmen, Guangdong | 112.2234 | 22.33324 | 1 |
| ZQSH | ZQ01 | Sihui, Zhaoqing, Guangdong | 112.7064 | 23.33962 | 1 |
| GZTH | GZ01, GZ02, GZ03, GZ04, GZ05, GZ06, M1, M2, M3, M4, M5, M6, T1, T2, T3 | Tianhe, Guangzhou, Guangdong | 113.876 | 23.30645 | 15 |
| GZZC | GZ07, GZ08, GZ09 | Zengcheng, Guangzhou, Guangdong | 113.876 | 23.30645 | 3 |
| HZBL | HZ01, HZ02, HZ03, HZ04, HZ05 | Boluo, Huizhou, Guangdong | 114.2419 | 23.20189 | 5 |
| HZHD | HD01 | Huidong, Huizhou, Guangdong | 114.7853 | 23.18096 | 1 |
| CZRP | RP01 | Raoping, Chaozhou, Guangdong | 116.9247 | 23.75595 | 1 |

### 2.2. Nodule Surface Sterilisation

For further analysis, the samples were sterilized on their surfaces, frozen with liquid nitrogen, and kept at −80 °C. Root nodules that had been sterilized on their surfaces were treated for 30 s with 75% alcohol and for 8 min with 0.1% mercury bichloride ($HgCl_2$), then with sterilized water for five treatments. Sterilization of the surface was verified by water smearing following the fifth wash onto prepared yeast extract peptone dextrose (YEPD) and potato dextrose agar (PDA) plates.

### 2.3. 16S rRNA Gene Amplification and PacBio Sequencing

Using the TGuide S96 Magnetic Soil/Stool DNA Kit (catalog number: DP712; Tiangen Biotech (Beijing) Co., Ltd., Beijing, China), total genomic DNA was extracted from the root nodules of *F. falcata* samples. The extracted DNA was evaluated for quantity and quality by electrophoresis on a 1.8% agarose gel, and purity and DNA concentration were identified using a NanoDrop 2000 UV-Vis spectrophotometer (Thermo Scientific, Wilmington, NC, USA). With primer pairs 27F: AGRGTTTGATYNTGGCTCAG and 1492R: TASGGHTACCTTGTTASGACTT, the full-length 16S rRNA genes (V1–V9) were amplified [26]. For multiplexed sequencing, using sample-specific PacBio barcode sequences, the forward and reverse 16S primers were appended. We selected barcoded primers since they reduce the formation of chimeras in contrast to the alternative protocol, which entails adding primers in a subsequent PCR reaction. The KOD One PCR Master Mix (catalog number: KMM-101; TOYOBOLife Science, Osaka, Japan) was utilized for PCR amplification (25 cycles), beginning with a 2 min denaturation at 95 °C, followed by 25 cycles of 10 s denaturation at 98 °C, annealing for 30 s at 55 °C, extension for 1 min and 30 s at 72 °C, and a final step of 2 min at 72 °C. Purified PCR amplicons were quantified by means of the Qubit dsDNA HS Assay Kit and the Qubit 3.0 Fluorometer (catalog number: Q33230; Invitrogen, Thermo Fisher Scientific, Eugene, OR, USA). After individual quantification, equal amounts of amplicons were combined. SMRTbell libraries were constructed using amplified DNA and the SMRTbell Express Template Prep Kit 2.0 (catalog number: PN 100-938-900; Pacific Biosciences, Menlo Park, CA, USA) in accordance with the manufacturer's instructions. On a PacBio Sequel II platform, purified SMRTbell libraries were sequenced using the 3Sequel II binding kit 2.0 (catalog number: PN: 101-789-500). Under the accession number PRJNA1009201, the bacterial sequences have been submitted to the NCBI.

### 2.4. Sequence Data Analyses

In order to obtain circular consensus sequencing (CCS) reads, the sequencing-generated raw reads were filtered and demultiplexed through SMRT Link (version 8.0; https://www.pacb.com/support/software-downloads/, accessed on 7 August 2023) with

minPasses $\geq$ 5 and minPredictedAccuracy $\geq$ 0.9. Using the PacBio Barcode Demultiplexer lima (version 1.7.0; https://lima.how/, accessed on 7 August 2023), the CCS sequences were then assigned to the corresponding samples. For the CCS acquisition analysis, the BMK Cloud Biological Cloud Computing Platform (BMK Cloud; https://www.biocloud.net/, accessed on 7 August 2023) was utilized. CCS reads with no primers or beyond 1200–1650 bp were discarded through a Cutadapt (version 2.7; http://cutadapt.readthedocs.org/, accessed on 7 August 2023) quality control procedure by recognizing forward and reverse primers and quality filtering [27]. The UCHIME algorithm (v8.1; https://drive5.com/usearch/manual/, accessed on 7 August 2023) detected and removed chimera sequences in order to acquire clear readings [28]. To output an ASV, clean reads were performed on feature classification using DADA2 (version 1.20.0; https://benjjneb.github.io/dada2/, accessed on 7 August 2023) [29]. All samples with ASV counts below two were then filtered. The ASVs were taxonomically annotated in QIIME2 (version 2020.6.0; https://qiime2.org/, accessed on 7 August 2023) utilizing SILVA (release 138.1; https://www.arb-silva.de/documentation/release-138/, accessed on 7 August 2023) and a 70% threshold for confidence [30,31]. Total clean readings from each sample were used to determine the endophytic abundance. In contrast, the relative abundance (RA) of endophytic ASVs (or at a given taxonomy level, such as genus or phylum) was calculated as a percentage [32]. Alpha and beta diversity were determined using QIIME2 to assess the degree of similarity between microbial communities from various locales. The alpha diversities (Shannon index) between different regions were compared using a column chart (at four of the nine sampled regions, samples were selected from representative trees with no biological duplications, Table 1). On the basis of the ASV tables, QIIME2 was utilized to calculate the rarefaction curves. Principal coordinate analysis (PCoA) was used to determine the beta diversity of ASVs detected in distinct regions. Utilizing Venn analysis, the co-distribution of endophytic ASVs in various regions was determined. The distributions of commonly shared core genera (c-genera) and c-ASVs in different regions were calculated and displayed by HeatMap of TBtools software [33], and the correlations between clearly classified genera (top20) in all samples were calculated and displayed by SparCC [34].

## 3. Results

### 3.1. Microbial Diversity and Microbiome Composition of the Root Nodules

Validation was performed on the endophytic microbiome profiles of all root nodule samples (Supplementary Table S1, Figure 2a). From all samples, 383,482 16S rRNA clean reads and 6973 ASVs were extracted. Using the Shannon index, alpha diversity analysis revealed that the MMDB group had substantially ($p < 0.05$) greater taxonomic diversity than the GZTH group (Figure 2b). PCoA revealed the differences in the bacterial communities among different groups. In addition, the bacterial endophyte community of the YJYC group was resolved from other samples in the PCoA plots. Notably, the bacterial endophyte communities of the HZHD group clustered with those of the MMDB group. The bacterial endophyte communities of the HZBL, CZRP, and GZTH groups are closely clustered together at the genus level (Figure 2c).

The bacterial endophytes detected in the root nodules of all the groups mainly belong to the phylum Proteobacteria (RA: 42–90%). In total, 29 bacterial phyla and 575 bacterial genera were detected in all samples. Among these, *Bradyrhizobium* was the most dominant genus (RA: 13–37%) in all the groups except for HZHD and GZTH groups. In the HZHD group, *Puia* was the most dominant genus (RA = 14%), while in the GZTH group, *Paucibacter* was the most dominant genus (RA = 34%). In addition, *Pseudomonas*, *Rhizobium*, and *Caballeronia* were the dominant genera (RA: 5–34%) in all the groups (Table 2). At the genus level, the CZRP, HZBL, GZTH, and JMEP groups were clustered together, likely due to an increase in the relative abundance of *Paucibacter* (Figure 2e). Additionally, at the ASV level, ASV2 (*Paucibacter*) and ASV72 (*Bradyrhizobium*) were dominant in CZRP group, as well as the HZBL, GZTH, and JMEP groups (Figure 2f).

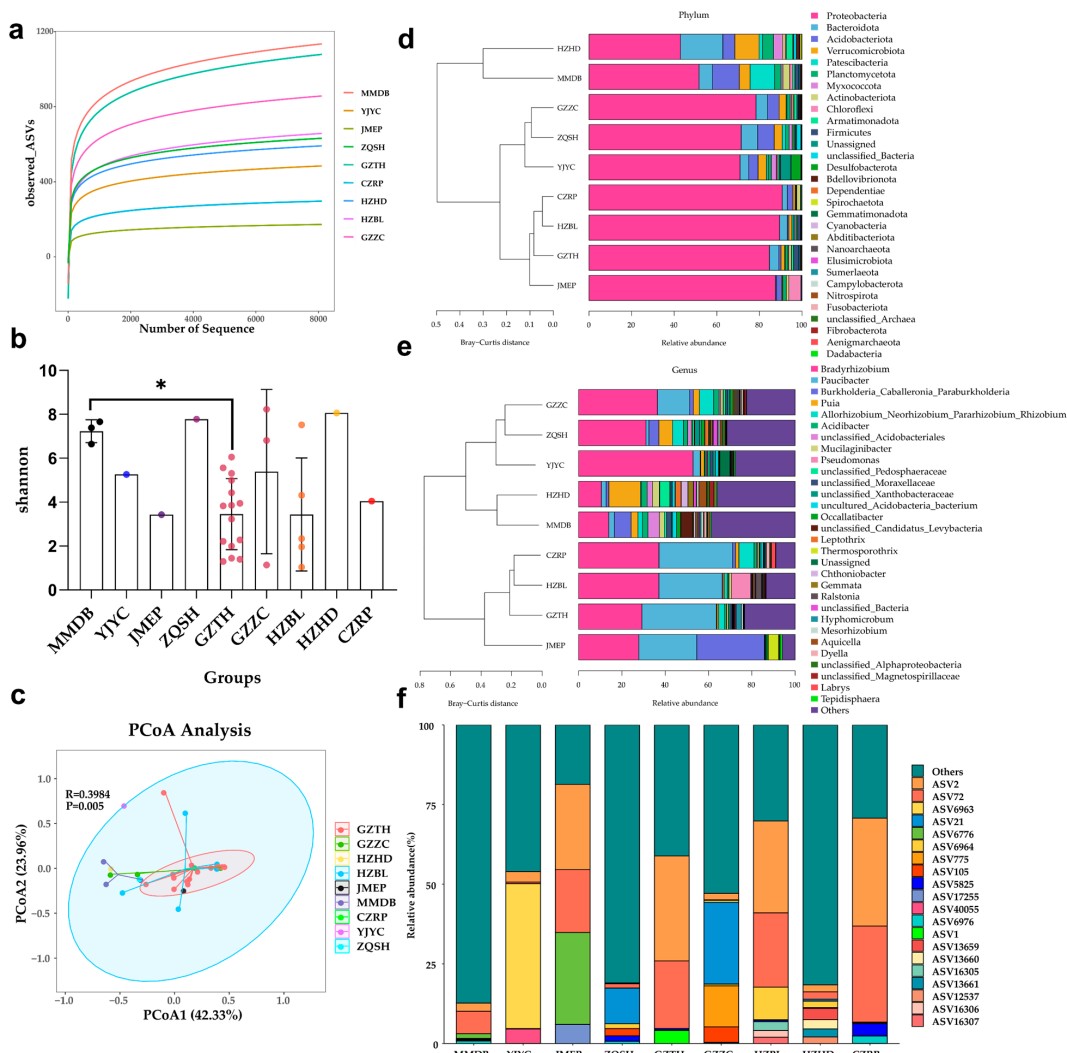

**Figure 2.** Diversity distribution of the root nodule bacterial endophytes from different groups. (**a**) The observed amplicon sequence variants (ASVs) rarefaction curves for 16S rRNA sequencing. (**b**) Shannon index of root nodule bacterial endophytes from different geographic groups (*: significance at the 0.05 level, $p < 0.05$). (**c**) PCoA profile of the endophytes in root nodules of different groups (for each axis, the percentage of variation is explained in square brackets). (**d**) Compositional and clustering analyses of the 30 most dominant endophytes at the phylum level in root nodules of different groups. (**e**) Compositional and clustering analyses of the 30 most dominant endophytes at the genus level in root nodules of different groups. (**f**) Compositional analysis of the 20 most dominant (RA > 2%) endophytes at the ASV level in root nodules of different groups.

**Table 2.** Relative abundance of dominant genera in all groups (*: RA > 5%; **: RA > 10%; ***: RA > 30%).

| Sample | Relative Abundance (%) | | | | | |
| --- | --- | --- | --- | --- | --- | --- |
| | *Bradyrhizobium* | *Paucibacter* | *Pseudomonas* | *Rhizobium* | *Puia* | *Caballeronia* |
| MMDB | 13.75 ** | 2.63 | 0.00 | 1.99 | 3.13 | 2.39 |
| YJYC | 52.40 *** | 3.30 | 0.00 | 0.17 | 1.66 | 0.00 |
| JMEP | 27.72 ** | 26.79 ** | 0.00 | 0.09 | 0.00 | 28.92 ** |
| ZQSH | 29.69 ** | 1.36 | 0.07 | 4.82 | 6.25 * | 0.37 |
| GZTH | 28.91 ** | 34.44 *** | 0.17 | 1.36 | 0.78 | 0.02 |
| GZZC | 36.33 *** | 14.89 ** | 0.03 | 5.69 * | 2.81 | 0.06 |
| HZBL | 36.95 *** | 29.18 ** | 9.01 * | 1.15 | 0.66 | 0.02 |
| HZHD | 10.50 ** | 2.23 | 0.18 | 0.29 | 14.59 ** | 0.10 |
| CZRP | 37.08 *** | 34.07 *** | 0.00 | 6.77 * | 1.54 | 0.04 |

### 3.2. The Dominant Endophytic Rhizobia for F. falcata Root Nodules

There was no statistically significant correlation between the abundance of bacterial endophytes and the number of ASVs in the analyzed root nodules. The CZRP cohort had the least amount of ASVs but the maximum number overall. In contrast, the JMEP group had the lowest number of ASVs but the third-highest abundance. Finally, the GZTH group had the highest number of ASVs and the second-highest abundance, after the CZRP group. However, it is important to note that over 99% of these ASVs had relative abundances of less than 1% in all the samples. This means that most bacterial endophytes in the root nodules had low relative abundance, except for a few ASVs that were responsible for the dominant relative abundance. Therefore, a large quantity of ASVs in the GZTH group is not inherently indicative of a greater overall abundance of bacterial endophytes in root nodules (Figures 2f and 3a).

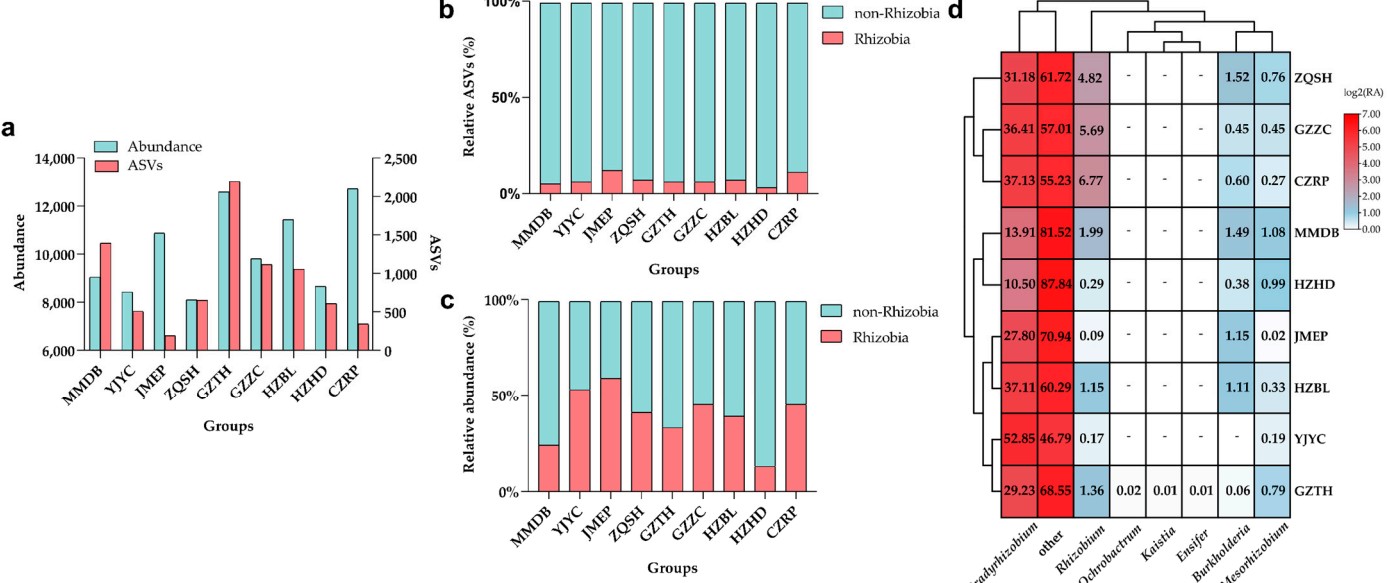

**Figure 3.** Comparative analysis of the endophytic rhizobia ASVs in terms of numbers and abundance in different groups. (**a**) The number and abundance of endophytic bacteria in root nodules in different groups; the number proportion (**b**) and relative abundance (**c**) of edophytic rhizobia (*Burkholderia*, *Rhizobium*, *Bradyrhizobium*, *Mesorhizobium*, *Kaistia*, *Ensifer* and *Ochrobactrum* genera) ASVs in root nodules in different groups; (**d**) sectional distribution of endophytic rhizobia in different groups.

While the number of endophytic rhizobia ASVs in all groups constituted a small portion (3.9–12.5%), the relative abundance of the endophytic rhizobia was dominant (RA: 46.8–87.8%) in the root nodules (Figure 3b,c). *Bradyrhizobium* exhibited the highest abundance (RA: 13.9–52.8%) in all groups among all the endophytic rhizobia. *Burkholderia*, *Bradyrhizobium*, *Mesorhizobium*, *Kaistia*, *Ensifer*, and *Ochrobactrum* accounted for only a small fraction (RA < 2%) of the overall abundance of endophytic rhizobia in all groups. Overall, *Rhizobium* emerged as the second most dominant rhizobium in all groups. However, its relative abundance in the JMEP and HZHD groups was lower than that of *Burkholderia* (Figure 3d).

A total of 18 c-genera and 2 c-ASVs were detected in nine groups (Figure 4a,b). The distribution pattern of endophytism among c-genera and c-ASVs showed a region-specific trend (Figure 4c,d). Notably, *Bradyrhizobium* and *Paucibacter* were the dominant genera within the c-genera. Interestingly, the relative abundance of *Paucibacter* was less than 4% in the MMDB, ZQSH, YJYC, and HZHD groups, while exceeding 26% in the JMEP, GZTH, HZBL, and CZRP groups (Figure 4c). Among the top 10 ASVs in terms of relative abundances across all samples, several c-ASVs were identified. These c-ASVs belonged to

the genera *Paucibacter* (ASV2), *Bradyrhizobium* (ASV72, ASV6946, ASV6964, ASV5, ASV21, ASV105, and ASV6782), *Rhizobium* (ASV5825), and *Burkholderia* (ASV6776). Notably, among the bacterial endophytes, c-ASVs, ASV2, and ASV72 were detected in all nine groups. The relative abundances of ASV2 and ASV72 were higher (RA: 19.7–33.9%) in JMEP, HZBL, GZTH, and CZRP groups, while they were lower (RA: 0.2–7.0%) in ZQSH, GZZC, MMDB, YJYC, and HZHD groups.

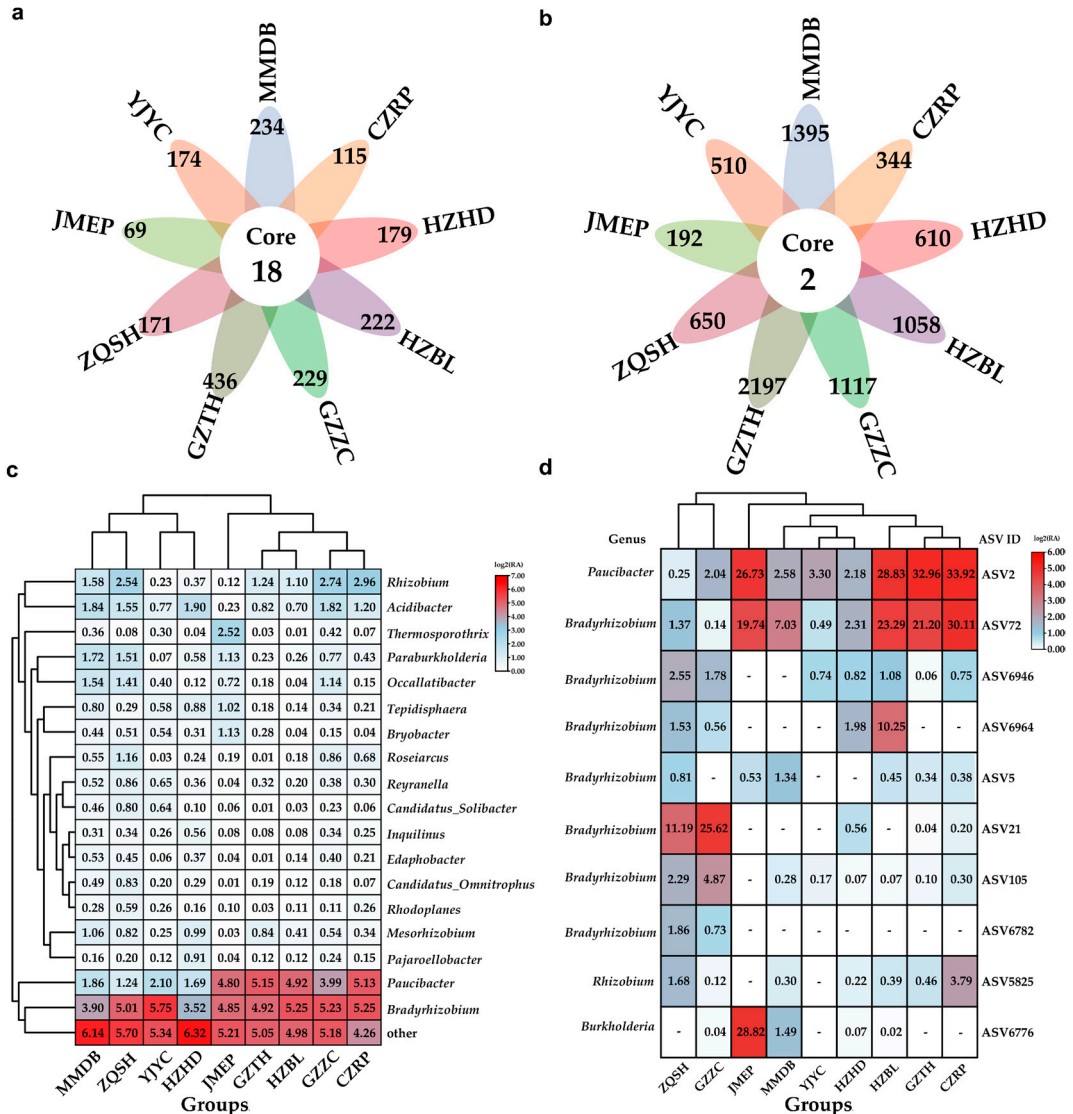

**Figure 4.** Distribution of commonly shared core genera (c-genera) and commonly shared core ASVs (c-ASVs) in root nodules of different groups. Venn diagrams exhibit the numbers of core-shared bacterial endophytes genera (**a**) and ASVs (**b**) in different groups. Heatmap depicting the sectional distribution of c-genera (**c**) and c-ASVs (**d**) in different groups.

### 3.3. Limited Symbiotic Correlations between Rhizobia and Other Endophytic Bacteria in Root Nodules

The correlation analysis of the classified genera (top 20) revealed that there were limited symbiotic correlations between rhizobia and other endophytic bacteria in root nodules. Specifically, *Bradyrhizobium* and *Paucibacter*, the dominant members of the root nodule endophytic bacteria community, did not show significant correlations ($p > 0.05$) with other endophytic bacteria. It was discovered that *Rhizobium* and *Burkholderia* have a significant positive correlation ($p < 0.05$) with *Dyella*. Furthermore, *Burkholderia* had a

robust positive correlation ($p < 0.01$) with *Caballeronia*, whereas *Rhizobium* had a substantial negative correlation ($p < 0.05$) with *Aquicella*.

## 4. Discussion

The 16S rRNA in the genomic DNA of 31 samples from Guangdong in southern China was sequenced and analyzed. In four of these groups, we exclusively obtained a single sample from the local representative tree (Table 1).

On the basis of their composition, we characterized the bacterial endophyte communities on *F. falcata* root nodules. Proteobacteria (RA: 42–90%) exhibited an overwhelming dominance across all groups (Figure 2d). Notably, endophytes of all groups could hardly be resolved in the PCoA plots (Figure 2c). Rhizobia bacteria were dominant in the root nodule microbiome, similar to other legumes [35]. Interestingly, the relative abundance of rhizobia, which accounted for only a small proportion (3.9–12.5%) of ASVs, ranked first (RA: 46.8–87.8%) in all samples. This finding suggests that bacteria endophytes other than rhizobia are present in nodules at a low relative abundance.

*Bradyrhizobium* was much more abundant in root nodules than other bacteria endophytes with nitrogen-fixing effects (such as *Burkholderia*, *Rhizobium*, *Mesorhizobium*, *Kaistia*, *Ensifer*, and *Ochrobactrum*) (Figure 3d). In the c-genus, *Bradyrhizobium* exhibited a remarkable dominance in the root nodules of *F. falcata* in all groups (Figure 4c), which aligns with the findings from the isolation of *F. falcata* in Indonesia [17]. Additionally, *Bradyrhizobium* displayed a high dominance in northern China [36], eastern North America [37] and Australia [38] in legumes, where it formed nodules on legumes adapted to acidic soils. The currently available evidence suggests that *F. falcata* prefers *Bradyrhizobium*. *Bradyrhizobium* is also the favored symbiont of other legumes [39,40] and nonlegumes (*Parasponia*) [41]. Its ability to form symbiotic relationships with the nodules of numerous plants may be attributed to the high adaptability of *Bradyrhizobium* in terms of hosts. Considering this characteristic, *Bradyrhizobium* may facilitate *F. falcata*'s rapid adaptation to new soil conditions. This unique ability may facilitate the rapid adaptation of *F. falcata*, an exotic tree species, to the Chinese environment. *Bradyrhizobium* was once considered the first symbiosis-compatible Proteobacteria, compatible with legumes after their introduction into a new environment [6]. However, it is worth noting that other rhizobia apart from *Bradyrhizobium*, such as *Burkholderia*, *Rhizobium*, and *Mesorhizobium*, have been detected in *F. falcata* root nodules from nine groups in southern China. This raised uncertainty regarding whether *F. falcata* was indeed the first plant species to establish a symbiotic relationship with *Bradyrhizobium* and form root nodules. Nonetheless, *Bradyrhizobium* remains a core bacterial endophyte that adapts to *F. falcata* root nodules in southern China.

The values of diversity indices (Shannon) indicate the existence of more bacterial diversity in different groups (Figure 2b). In addition to rhizobia, the main endophytic bacteria found in *F. falcata* root nodules include *Paucibacter*, *Puia*, *Pseudomonas*, and *Caballeronia*. It is important to highlight that in a specific group, *Paucibacter*, *Puia*, and *Caballeronia* exhibited the highest relative abundance, surpassing even *Bradyrhizobium*, in *F. falcata* nodules (Table 2). Interestingly, no significant correlations were observed among the top 20 genera in all samples, suggesting that the composition of nodule bacteria may be closely associated with the rhizosphere [42] and potentially influenced by the strain-specific association with *F. falcata* [7]. This finding may explain the wide variation in the relative abundance of c-genus and c-ASVs, such as *Paucibacter*, ASV2 (*Paucibacter*), and ASV72 (*Bradyrhizobium*), observed across different groups (Figure 4c,d). Endophytes were monitored at the ASV level as opposed to other taxonomic levels (such as genus or species) in order to assure the consistency of endophytes detected across various groups as opposed to the same species or genera. At both the genus and ASV levels, *Paucibacter* was consistently detected in all groups. Despite being a nonrhizobia with uncertain specific functions, *Paucibacter* was identified as a core bacterial endophyte in this study due to its consistent detection across all groups and its dominant presence in certain groups. The outcomes indicate that the bacterial community structure within the root nodules of *F. falcata* demonstrates plasticity.

Studies concerning the diversity of nodule bacteria and identifying the core rhizobia that inhabit *F. falcata* root nodules have agricultural and horticultural importance. The majority of plant-associated bacteria, rhizobia in particular, are located in soil [43]. However, physiological and environmental changes frequently reduce plant surface colonization, and only adapted bacteria can survive and penetrate the plant via wounds, lesions, and hydathodes [13]. Given these circumstances, the inoculation of specific endophytic bacteria becomes particularly interesting. It is widely recognized that the deliberate inoculation of specific endophytic bacteria can effectively enhance plant growth and improve the overall structure of the flora [13]. This suggests a possible strategy of purposely inoculating with certain beneficial rhizobia or bacteria to help plants rapidly form nodules and adapt to their new environment. Identifying the core rhizobia that inhabit *F. falcata* root nodules can greatly help in conducting the targeted inoculation of endophytic bacteria.

Figure 5 indicates that the absence of symbiotic relationships between rhizobia and other endophytic bacteria in root nodules suggests that the external environment influences the structure of bacterial endophytes. The rhizosphere is widely recognized as a complex and dynamic environment [44], and it is highly likely that the rhizosphere of *F. falcata* in different regions of southern China exhibits significant variation. It is important to observe, however, that we did not obtain distinct soil samples from the rhizosphere. Therefore, we were unable to ascertain whether *Paucibacter* and *Bradyrhizobium* were selectively enriched in the rhizosphere, or inherently abundant in our soils relative to other taxa.

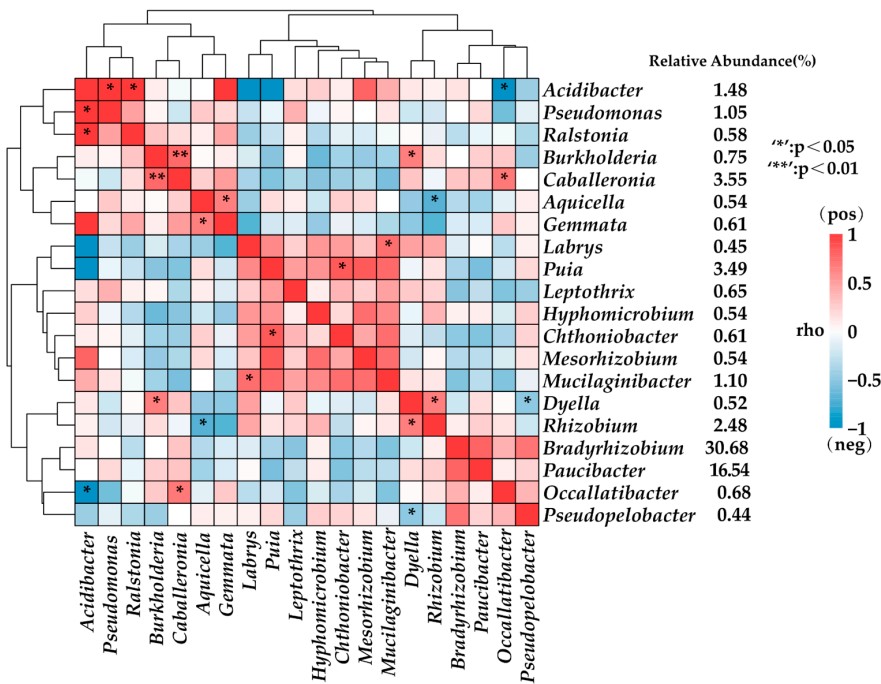

**Figure 5.** Symbiotic relationship of the bacterial endophyte community in root nodules.

## 5. Conclusions

This study examined the structure of bacterial endophytes in the root nodules of *F. falcata* after its inception in southern China. This study successfully identified *Bradyrhizobium* and *Paucibacter* as the core bacterial endophytes inhabiting *F. falcata* root nodules at the genus level. In addition to rhizobia, in nodules of *F. falcata*, *Paucibacter* was the most common genus detected. However, its specific role within the nodules is still not well understood. Overall, the structure of bacterial endophytes within the root nodules of *F. falcata* could be influenced by variations in geographic location. Nevertheless, specific bacterial endophytes exhibited a strong affinity for *F. falcata* and could be found across different regions. These bacterial endophytes demonstrated a remarkable specificity toward *F. falcata,* and likely contributed significantly to the nodulation process of *F. falcata*.

**Supplementary Materials:** The following supporting information can be downloaded at: https://www.mdpi.com/article/10.3390/d15101092/s1, Table S1: Statistics of 16S rRNA amplicon sequences in root-nodule samples of different groups.

**Author Contributions:** Conceptualization, S.Y. and H.Z.; methodology, S.Y.; formal analysis, S.X. and S.Y.; investigation, Q.L., R.H., R.W. (Runhui Wang), R.W. (Ruping Wei) and G.W.; writing—original draft, S.X. and S.Y.; writing—review and editing, S.X. and H.Z. All authors have read and agreed to the published version of the manuscript.

**Funding:** This research was supported by the Specific Programs in Forestry Science and Technology Innovation of Guangdong (No. 2022KJCX007).

**Data Availability Statement:** All data and materials are available in the manuscript and Supporting Information, and in NCBI under the accession number PRJNA1009201.

**Conflicts of Interest:** The authors declare no conflict of interest.

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
