# Peer review of "Characterization of the Root Nodule Microbiome of the Exotic Tree Falcataria falcata (Fabaceae) in Guangdong, Southern China"

_diversity, doi:10.3390/d15101092_

Round 1

Reviewer 1 Report

The manuscript is well written, clear with clearly explained results and I have no major complaints except the following:

1. Fig. 2 d and c legend must be clear; the legend is not visible at all.

Perhaps the following should be discussed as well;

2. Is the Bradyrhisobium characteristic for the substrate on which F. falcata was planted, or is it just more numerous and easier to enter into symbiotic relations with the plant?

3. What affects the number of bacteria entering into symbiotic relationships?

I am not a native speaker but I did not have any problem to read and understand tha manuscript; maybe there are some minor language errors. I am not qualify for English.

Reviewer 2 Report

The manuscript entitled “Characterization of the Root Nodule Microbiome of the Exotic Tree Falcataria falcata in Guangdong, Southern China” deals with the 16S metagenomic analysis of root nodule endophytes of different groups from nine geographic regions in Southern China. The article is well written, and the scientific soundness is adequate for Diversity; however, the authors must address the following issues:

1.       Please add the family after the scientific name in the title (Fabaceae)

2.       Line 16. Please substitute “great alterations” for “differences”.

3.       Line 36. Please add “genera” after Ochrobactrum

4.       Line 37. Plant genetic factors are not mentioned. Please add.

5.       Line 42. Please add “relative abundance” after rhizobia

6.       Line 52 to 54. The idea on the sentence “Bradyrhizobium has been isolated from F. falcata root nodules in various regions in Indonesia [16]. However, it is important to note that this alone is not sufficient evidence to conclude that Bradyrhizobium is the core rhizobium of F. falcata root nodules.” Must be included in the discussion, not in the introduction. The idea must be mentioned when new evidence is faced with previous reports.

7.       Line 54. The phrase “The number of endophytic bacteria artificially isolated” is not clear. Do you mean to biased sampling or any other methodological procedure? Please clarify.

8.       Line 59. Please substitute “bacterial endophytes” for “bacterial endophyte communities.”

9.       Line 66. Please substitute “it is crucial to understand that” for “the”.

10.   Figure 1 is not adequate for the context of the work. Please substitute the image for a GIS map showing a layer of forest integrity or vegetation type coverage. Longitude scale is wrong, it starts at 109° E, then goes in 0.5° to 111.5°, then jumps to 120° ending at 125°. That’s in the ocean. Please correct as follows: Longitude starting in 110°, ending in 118° E increasing in 2°. For latitude scale please show only 21 to 24° N in 1° increases. Font and size in figure labels must fit the journal’s style. Scale in Km is not visible.

11.   Table 1. Headings in latitude and longitude are inverted.

12.   Table 1. Column6. Please substitute “sampling” for “sampled”.

13.   Lines 98, 108, 112, 116, 119. Please add catalog number for employed kits.

14.   Line 104. Please cite the primers report. Please state the V target region of selected primers.

15.   Line 123. Please cite SMRT Link software, add url.

16.   Line 125. Please add “The PacBio Barcode Demultiplexer” before “lima”. Please cite, add url.

17.   Line 128. Please cite Cutadapt , add url.

18.   Line 129. Please cite UCHIME, add url.

19.   Line 131. Please cite DADA2, add url.

20.   Line 133. Please cite QIIME2, add url.

21.   Line 133. Please cite SILVA, add url.

22.   Line 134. Please state the reason why 70% confidence was used (see https://journals.asm.org/doi/full/10.1128/msystems.00518-21)

23.   Line 135. Please substitute “one” for “each”.

24.   Line 137. Please substitute” other” for “a given”

25.   Line 137. “level” (singular)

26.   Line 137. Add “or” after “genus”

27.   Line 140. Please add “sampled” after “nine”

28.   Line 153. Please continue in a single paragraph.

29.   Line 155 and further. Please add “group” or “groups” after each mention of MMDB GZTH, HZHD, CZRP, etc.

30.   Figure 2 is mostly unable to read. Please modify all labels to fit font and size of footnote (Palatino linotype, 11 points).

31.   Line 172. Please substitute “bacterial endophytes in different regions of root nodules” for “root nodule bacterial endophytes from different groups”

32.   Line 174. Please substitute “root-nodule endophytes in different regions” for “root-nodule bacterial endophytes from different geographic groups”

33.   Line 174. Undercase p in (P<0.05)

34.   Table 2. Please remove. Include as Supplementary File.

35.   Table 3. Please remove “_” before Caballeronia

36.   Table 3. Please make sure values are aligned at the decimal point.

37.   Line 185. Please substitute “these” for “analyzed”.

38.   Please substitute “regions” by “groups” in results and discussion sections

39.   Figure 3. Please edit all labels to fit font and size (Palatino linotype 11 pts), except subfigure label (a, b, c, d)

40.   Figure 3a. ASV must appear also as bars, not line.

41.   Figure 3a, b and c. Please add “Groups” as label for horizontal axis.

42.   Figure 3d. Please italicize taxonomic names in horizontal axis labels

43.   Line 218. Please move “of” before “(c)”.

44.   Line 219. Please add “genera” after Ochrobactrum

45.   Figure 4. Please edit all labels to fit font and size (Palatino linotype 11 pts), except subfigure label (a, b, c, d)

46.   Figure 4c and d. Please add the scales for distances in clusters for horizontal and vertical axes. Add “Groups” as label for horizontal axes.

47.   Figure 4d. Please italicize taxonomic names.

48.   Lines 232, 233, and 234. Please undercase the letter p in probabilities.

49.   Figure 5. Please edit all labels to fit font and size (Palatino linotype 11 pts). Undercase letter p in probability

50.   Line 296. Please substitute “crucial” for “interesting”.

51.   Line 309. Please substitute “bacteria” for “taxa”.

52.   Line 311. Please remove “To the best of our knowledge, this study represented the first comprehensive report on the structure of bacterial endophytes within” Please add a sentence limited to introduce your results like “In this work we analyzed…”.

53.   Lines 333 to 419. Please add the DOI to each reference. Please italicize taxonomic names.

Reviewer 3 Report

The article raises an interesting issue. The authors have done a lot of work, however, a few questions and comments:

Comments

1. Figure 4 c, d - illegible, couldn't it be better to do it vertically ?

2. Figures 2a d e, 3 d - illegible

3. In which year was the research conducted? In one year? Two? no test date given

4) How many replicates were there within 1 tree, or just 1 pooled sample of all from the tree taken root noduls

5. Table 2. statistics of 16S rRNA amplicon sequences in root-nodule samples of different regions.- what test, the beginning of the table caption mentions statistic

6. Please read the MS  once more and correct any minor shortcomings, e.g. punctuation, etc.

Round 2

Reviewer 3 Report

The authors have made all recommended corrections. In addition, the authors have also responded to all comments.
In my opinion, the article can be accepted for publication.